# Damage Propagation by Cyclic Loading in Drilled Carbon/Epoxy Plates

**DOI:** 10.3390/ma16072688

**Published:** 2023-03-28

**Authors:** Luis M. P. Durão, João E. Matos, Nuno C. Loureiro, José L. Esteves, Susana C. F. Fernandes

**Affiliations:** 1ISEP, Instituto Politécnico do Porto, Rua Dr. António Bernardino de Almeida, 4249-015 Porto, Portugal; 2Associate Laboratory for Energy, Transports and Aerospace (LAETA-INEGI), Rua Dr. Roberto Frias 400, 4200-465 Porto, Portugal; 3ISVOUGA, Instituto Superior de Entre Douro e Vouga, 4520-181 Santa Maria da Feira, Portugal; 4INEGI Instituto de Ciência e Inovação em Engenharia Mecânica e Engenharia Industrial, 4200-465 Porto, Portugal; 5FEUP Faculdade de Engenharia da Universidade do Porto, 4200-465 Porto, Portugal

**Keywords:** bearing load, cyclic load, drilling damage, damage propagation

## Abstract

Fiber reinforced composites are widely used in the production of parts for load bearing structures. It is generally recognized that composites can be affected both by monotonic and cyclic loading. For assembly purposes, drilling is needed, but holes can act as stress concentration notches, leading to damage propagation and failure. In this work, a batch of carbon/epoxy plates is drilled by different drill geometries, while thrust force is monitored and the hole’s surrounding region is inspected. Based on radiographic images, the area and other features of the damaged region are computed for damage assessment. Finally, the specimens are subjected to Bearing Fatigue tests. Cyclic loading causes ovality of the holes and the loss of nearly 10% of the bearing net strength. These results can help to establish an association between the damaged region and the material’s fatigue resistance, as larger damage extension and deformation by cyclic stress contribute to the loss of load carrying capacity of parts.

## 1. Introduction

Carbon/epoxy composite laminates are widely used in the production of load-bearing primary structures due to their highly specific strength, stiffness, fatigue strength and impact resistance, thermal conductivity, corrosion resistance, and good dimensional stability [1] when compared with metallic alloys. Examples of the advantageous use of fiber reinforced composites can be found in several fields of technology, see [2,3,4,5]. During the production process and further industrial application of parts made from carbon/epoxy composite laminates, drilling circular holes may be necessary to allow their assembly and incorporation into more complex products, using screws, bolts, pins, rivets or snap springs. Other machining operations can also be performed due to concerns related to dimensional accuracy and good surface quality requirements [1].

These operations are currently carried out by machining, and drilling is still the machining operation most widely used and studied in a considerable number of published papers covering several types of composites. Ghabezi and Khoran [6] investigated the influence of cutting speed, feed rate, and tool diameter in composite sandwich structures with a PVC core. Ghabezi et al. [7] carried out an experimental study to characterize the delamination and uncut fiber in the drilling of honeycomb structures. Yasar et al. [8] and Rajkumar et al. [9] also evaluated the influence of cutting parameters on hole quality for carbon/epoxy composites. A common outcome of these studies was the recognition of the superior contribution of feed rate to delamination extension. Most recently, Jinyang Xu et al. [10] conducted an experimental study on drilling damage for woven GFRP composites, contributing to a better understanding of the composites machining process. A recent review on the mechanisms of delamination formation, assessment, and suppression during the drilling of composites can be found in Geng et al. [11].

Even though drilling has been widely studied in recent years, for the purpose of drilling parameters optimization and their outcomes in terms of damage around the hole, there are few studies that address the fatigue behavior of composites in the vicinity of the hole, namely on the propagation of the damage when the parts are subjected to cyclic loads.

Generally, it is accepted that in the drilling process, delamination is the most frequent cause of damage and can reduce the load-bearing capacity of the parts. One of the main problems is related to the abrasive nature of the reinforcement fiber, which can lead to rapid tool wear and deterioration of the hole machined surfaces. The most typical damages caused by the drilling operation are pushout or exit delamination, peel-up or entry delamination, several hole wall damages such as fiber pullout, burrs, splintering, swelling, as well as thermal damages [1,12,13,14], see Figure 1.

Considering the drilling of a composite part, the desired results can be achieved with a smart combination of drilling tools and cutting parameters, tailored to the material to be cut, and the material properties such as the interlaminar fracture toughness in Mode I [15]. When using a common drill, there is a small region around the center of the chisel edge where the tool does not cut but extrudes the material, called the indentation zone. Considering that fiber-reinforced composites are, in general, more brittle than metals, it is unlikely that extrusion takes place, and orthogonal cutting could be assumed for the entire chisel edge. Drill bit geometry plays an important part in thrust force and, therefore, in delamination onset and propagation. The chisel edge of a twist drill acts like a punch over the uncut plies of the laminate. The consequence of this action is delamination whenever this thrust force exceeds the interlaminar resistance of the laminate. The size of the delamination zone has been proven to be related with the thrust force developed during the drilling process [16]. The use of special drill bits can contribute to the use of larger feeds without delamination onset. Overall, it is accepted that a drilling process that is able to avoid delamination should keep thrust forces below a threshold value, and this value is dependent on the uncut thickness, known as the critical thrust force (Fcrit) [17], see Equation (1). This critical thrust force is associated with several properties of the unidirectional laminate, such as the elastic modulus, E_1_, the Poisson ratio, ν_12_, the interlaminar fracture toughness in mode I, G_Ic_, and the uncut plate thickness (h):(1)Fcrit=π[8∗GIc∗E∗h33(1−ϑ122)]12

The work of Hocheng and Tsao has contributed significantly to the understanding of the delamination mechanism associated with diverse drilling conditions such as drill geometry [18] or tool wear. The hole can be predrilled to eliminate the thrust caused by the chisel edge. Thus, delamination risk is significantly reduced by the minimization of the thrust force, thus reducing delamination hazard [16]. Durão et al. [19,20,21] have followed this tool concept and developed several comparative studies comparing step and conventional drills. In [22], Tsao defined that an optimal ratio of first to second stage drill equal to 0.4 should be used to minimize delamination. Pilot hole drilling can be a good alternative for the reduction of thrust force, thus delamination. The tool geometry concept is essential to achieve good results, demonstrating the importance of a dedicated geometry for the drilling of fiber-reinforced laminates.

Studies on the load bearing consequence of drilling damage are usually concerned with the monotonic loading of drilled parts and their strength, as it was believed that these materials were not sensitive to cyclic effects. More recently, it has been recognized that composites can be affected by cyclic loading, leading to the need to establish the grounds for fatigue behavior knowledge. In fact, the initial drilling operation can act as a stress concentration notch, as it is understood that some delamination is prone to occur, followed by damage propagation from that point on, leading to multifaceted damage progression sequences and, finally, failure. This notch sensitivity can be affected by several factors including, among others, the laminate thickness, ply orientation, laminate size, notch size, and machining quality. This quality is governed by the drilling operation parameters—cutting speed and feed rate—as well as by the drill material and by the drill bit geometry.

The first known study on this effect is the work of Persson et al. [23], which shows the consequences of different tools on the fatigue behavior of drilled parts. From this starting point, several published papers are available that try to establish a sound correlation between the damage and fatigue behavior of composite plates. The first approach to this problem was from the aeronautical field [24,25], stating that, unlike metals, fatigue in composites develops in stages, each distinguished by a set of cracks. Experimental investigations into the damage development and residual strengths of open-hole specimens in fatigue were conducted and a model developed [26,27,28]. In this work, the authors used a cyclic load at 5 kHz to investigate the sequence of damage events leading to the final failure of the plate. It is interesting to note that fatigue delamination progression was assessed via X-ray CT. Finally, as previously expected, the log (number of cycles to failure) decreased linearly as the maximum fatigue stress level increased [26]. At the end of a predetermined number of cycles, a quasi-static loading until failure was performed [27]. Authors concluded that the residual strengths are similar for each of the severities tested Other more recent studies based on experimental analysis focused on evaluating the effect of a notch or stress concentrations on the fatigue performance of laminated structures, such as [29,30]. Muc and Romanowicz [30] demonstrated that crack initiation and growth are related to the interaction of the stress intensity factors. Hochard et al. [29] presented a fiber failure model for static and fatigue loads but recognized that further studies are needed. Yenigun and Kilickap [31] investigated the effect of drilling quality on the fatigue life of unidirectional, cross-ply and ±45° fiber angle carbon fiber-reinforced plastics. It was observed that with the decrease in drilling quality, a significant decrease occurred in the tensile force and fatigue life of the laminates. Sypt et al. [32] analyzed the failure mechanism of CFRP composite laminate under a cyclic pin-bearing load in quasi-isotropic laminates. In this thorough experimental and numerical modeling study, the authors presented the effect of several factors such as the loading factor and the loading ratio with a frequency of 5 Hz. Rui Guo et al. [33] analyzed the effect of tension-tension fatigue in composite rods with uniformly dispersed hybrid carbon/glass fiber, comparing the enhanced performance with glass fiber shell/carbon fiber core hybrid rods, showing that fatigue life has improved for the former mode.

Some studies also model the prediction of the open-hole strength plates, as seen in the papers of Camanho et al. [34] and Bao and Liu [35], or the studies on the effect of compressive stresses, such as in [36,37], thus providing a comprehensive base for the work here presented.

In the work here presented, two batches of carbon/epoxy plates were drilled under different machining conditions regarding the drill geometry, see Figure 2, and feed rate. The difference between the two baches is the stacking sequence, as one batch was of unidirectional plates [0°]_24_ and the other was of cross-ply plates with the following stacking sequence [(0/90)_6_]_s_. During drilling, the thrust force was monitored to identify the likelihood of delamination onset. After drilling completion, the hole surrounding area was inspected by enhanced radiography, a non-destructive method combining a contrasting liquid penetrant with a digital image of the hole plus the surrounding delaminated edge. Radiographic images were obtained with the help of a 60 kV, 300 kHz Kodak 2100 X-ray system associated with a Kodak RVG 5100 digital acquisition system. From these images, the damaged area and other geometrical features of the damaged region were computed, using appropriate MatLab^®^ tools, such as the Image Processing Toolbox.

Subsequently, the test specimens were subjected to cyclic bearing forces in a Shimadzu AG-X 100 kN Universal Testing Machine, (Shimadzu Corporation, Kyoto, Japan) as described in ASTM Bearing Fatigue Response of Polymer Matrix Composite Laminates—ASTM D6873—19 [38]. In the work here presented, specimens were not loaded until failure, as the test was halted after each thousandth cycle. Then, a new non-destructive inspection was carried out, and damage propagation extension was determined by comparing consecutive digital images by means of the image processing tools described above. This sequence was repeated twice for as many coupons as possible, providing that final failure did not occur, although failure did occur for a few. Finally, specimens that did not break under cyclic loading were bearing loaded until failure for residual strength assessment.

This experimental sequence has been presented in past papers from the same research team, see [39,40]. The only significant difference is the loading cycle, which is programmed with the help of the Trapezium software available with the universal testing machine.

The results of the experimental sequence here presented allow us to identify some differences in relation to the stacking sequences when under cyclic loads and, also, to have some information on the effects of cyclic loading on drilled composite plates. On the ither hand, cross-ply plates were able to withstand some deformation, showing some ovality of the hole, while most of the unidirectional plates failed after a reduced number of cycles. For the cross-ply plates, it was possible to draw a fatigue curve showing that a correlation between number of cycles and load amplitude can eventually be established. There is a decrease in the load bearing capacity of all the plates as the number of loading cycles increase. On the other hand, the load amplitude of the tests to unidirectional plates was always inferior to that of the cross-ply plates.

## 2. Materials and Methods

Fatigue testing is time and material consuming. The experimental work predicted needed to be adapted to keep the time frame within a reasonable period, as the main target was to have a significative number of results that could enhance further steps on the development of this type of material damage evaluation. Therefore, two batches of carbon/epoxy plates were produced: cross-ply and unidirectional. The test plates were produced from CIT (Composite Materials Italy, https://www.composite-materials.it/pagina.php?cod=1, accessed on 24 September 2022) carbon prepreg “CIT HS160 T700 ER450 UD tape 36%” and cured in an autoclave for one hour under 300 kPa and 130 °C, followed by cooling. The cross-ply test plates consisted of 24 layers of unidirectional prepreg, stacked in a symmetric cross-ply sequence, alternating 90° between each layer, corresponding to a stacking sequence of [(0/90)_6_]_s_, and coupons were supplied with the dimensions of 153 × 35 × 3.6 mm^3^. The unidirectional plates had the same number of layers but all of them follow the same alignment, with all the fibers oriented towards the main load direction. Plate mechanical properties can be found in Table 1.

The experimental work began with the drilling of the laminate plates for thrust force monitoring, delamination measurement by enhanced radiography, automated computational algorithms of image processing and analysis and, finally, mechanical tests.

The drilling operation was performed in a HAAS VF-2 CNC, while thrust forces were monitored with the help of a Kistler 9171A load cell. As has been previously identified, feed rate is crucial compared to spindle speed in the development of thrust forces [42]. The cutting speed and feed rate were kept constant and equal to 2650 rpm and to 0.05 mm/rev, respectively. These cutting parameters were selected according to previous published works [20,42,43], keeping thrust forces below the value of the critical thrust force as well as the tool manufacturer’s recommendation. When considering a layer thickness of 0.15 mm, F_crit_ is equal to 67 N, by Equation (1). A tool diameter of 6 mm was used combined with variations in drill geometry—Brad and twist (see Figure 2)—and material, as the latter were of tungsten carbide (WC) and of high-speed steel (HSS). BRAD drill is a commercial drill normally available in the tool manufacturers catalogue, and the twist drill is still the most common drilling tool geometry used in almost every tool shop. The standard option is a carbide drill, identified in this work as TWIST WC, but a TWIST HSS was also included with the intention to create a “bad” reference set, as normally this is considered a type of drill material to avoid when drilling of CFRPs is concerned.

After drilling, it is necessary to quantify the delamination extension caused by the machining operation. Normally, non-destructive tests (NDT) are used for this evaluation. For the damage assessment in this work, coupons were immersed in diiodomethane, a contrasting liquid, for 15 min and then radiographed with the help of a digital imaging system consisting of a 60 kV, 300 kHz Kodak 2100 X-ray system associated with a Kodak RVG 5100 digital acquisition system. The exposition time was set to 0.25 s. Once the digital images are acquired, it is possible to proceed with the necessary image processing to identify the damaged region and to compute the required parameters regarding the areas or diameters both the drilled hole and damaged area. The final purpose of the image processing and analysis sequence is the calculation of the assessment factor selected for this study, the Delamination factor (F_d_), as in Bajpai et al. [44], Equation (2), where A_max_ is the maximum area (hole area + damage area) and A_hole_ is the geometrical value of a circle with the nominal diameter of the drill. An analogous experimental method was already implemented and used by the authors with interesting results [45].
(2)$$F_{d}=\frac{A_{MAX}}{A_{hole}}$$

Further information on damage assessment criteria can be found in Geng et al. [11].

The next step was the completion of the mechanical tests. First, a few coupons were tested according to ASTM D5961-17 [46] Procedure A, both for unidirectional and cross-ply plates. This preliminary test was intended to set a strength value corresponding to the maximum bearing load, or 100%, that the material can bear in a quasi-static test. After this reference value was set, the plates were tested under different cyclic loadings at a low frequency, monitoring the mechanical response of the pin-loaded CRFP laminates until final failure of the plate or until a predefined number of cycles is reached. The test setup is in accordance with ASTM D6873 [38]. Due to the characteristics of the equipment used, it was decided to stop after 1000 cycles, corresponding to approximately 6 h of testing, or when pin displacement reached 0.6 mm, corresponding to 1/10 of the hole diameter. If failure occurs, the test was interrupted. Then, the plates were radiographed to observe and assess damage propagation caused by cyclic loading and then a new cyclic loading test was repeated. The maximum stresses used along this experimental phase are presented in Table 2, considering a loading ratio R = 0.1, as found in similar studies [26], to define the minimum stress. All the values of the fatigue testing sequence were rounded up for convenience.

Finally, all the plates without failure during cyclic tests were bearing tested according to ASTM D5961 [46] to check for residual strength. Results of this complete experimental sequence are presented in the following section.

## 3. Results and Discussion

### 3.1. Thrust Force Monitoring

The most significant value considered in this work was the axial thrust force (F_z_) during drilling, as normally the main issue reported is the need to keep thrust force below the Critical Thrust Force (see Equation (1)) to avoid damage onset and propagation. Although the data collected include the forces in the plate plane (F_x_, F_y_), as well as torque, the range of values is too low to be regarded as influential in any outcome.

The results, presented for cross-ply plates only, include the typical development of thrust force curves along the drilling operation, see Table 3 and Figure 3. From the drilling data, it was also possible to compute the thrust force value when the uncut thickness was equal to 0.15 mm. This last result is comparable with the value of F_crit_ determined in the previous section.

From the results available, it is not only possible to affirm that these curves are mainly influenced by drill geometry, but also that the maximum thrust force value or thrust force at a certain path registered during the drilling process is a function of drill geometry, with the plates being identical. Thus, when using the Brad drill, thrust force values were always lower than those observed with the Twist WC drill. The only possible reason for this is the drill geometry itself, as the twist drill has a higher piercing action exerted by the drill tip than the Brad drill design, which enables a pre-tensioning of the fibers before being cut. The result is a clean cut and a smoother cutting action, resulting in less roughness of the hole walls, an effect not included in this work, and, eventually, less delamination. When using Twist HSS drills, the thrust force values were always higher, confirming the established difficulty to machine CFRPs with HSS tools, leading to bad finishing and higher delamination around the machined hole.

### 3.2. Damage Assessment and Bearing Strength

After the drilling phase was completed, the next step was the radiography of all the plates drilled for a complete mapping of the damaged region around the hole. The results, considering the damage criteria defined in Equation (2), are presented in Table 4 as the average values for each drilling condition, including stacking sequence and drill geometry.

One of the motivations for this experimental work is that there should be some correlation between the damage extension, here assessed by the Delamination factor and the mechanical resistance of the plate that can be measured by the Bearing Strength. In Figure 4, the results of this correlation for the cross-ply plates are presented. Note that, even though there is some uncertainty in the results as is usual in composites, there is a clear trend on the cutback of the bearing strength as the damage extension is larger. Therefore, it is possible to affirm that a larger extension of the delaminated area, meaning a reduction in the consolidated resistant area in the vicinity of the hole, will cause a reduction in the bearing strength capacity of the plates. This outcome was expected and can be explained by the fact that a larger delamination extension will act as a pre-existing crack. As the crack is wider, the propagation of delamination makes the reduction in the cross-section resistance area easier.

On the other hand, when carrying out the same analysis for the unidirectional plates, it is possible to conclude that no correlation is possible, as is easy to conclude just by observing the values in Table 4, where the higher bearing strength was obtained with the plates drilled with the HSS drill. In fact, the drilling damage can easily propagate in a direction parallel to fibers (0° direction), leading to the fragile fracture of the coupon, as demonstrated in Figure 5, showing a radiography of a unidirectional plate after drilling and completion of the bearing test. The longitudinal slit is clearly visible. This diverse outcome for cross-ply plates may be because the 90° aligned fibers in these plates will act as barriers to the damage, preventing its propagation. In fact, it is known from Fracture Mechanics that the existence of fibers perpendicular to the direction of crack propagation creates a fiber pulling mechanism, which results in augmented energy dissipation retarding crack expansion. Therefore, cross-ply plates are helpful in preventing the sudden propagation of existing damages in the plate when compared with unidirectional plates.

### 3.3. Cyclic Tests

The focus of this experimental study was the collection of basic information on the consequences of cyclic loading on bolted connections. This loading is more likely to represent the service conditions of composite plates, simulating the assembling in complex sets and the resultant loading. For that, it is necessary to define how to assess the consequences of cyclic loading. Plates are tested according to ASTM D6873 [38] for a limited number of cycles and then radiographed, making it possible to compare the damage progression along the loading direction, as represented in Figure 6. The hypothesis is the ovality of the delaminated area due to the forced deformation of the hole. Values of loads for cyclic tests are presented in Table 2. This hypothesis was confirmed for cross-ply plates, see Figure 7, showing the sequence of images for a drilled plate after drilling and then after 1000 and 2000 cycles. As stated before, due to operative conditions, each test lasted for approximately 6 h, leading to this selection of the number of cycles. The results of this assessment for the ovality of holes are presented in Table 5. No fracture occurred during these tests for the number of cycles performed.

For unidirectional plates, ovality was not so relevant; the ovality measured in L2 just differed 0.04 mm from zero to 2000 cycles; however, some plates fractured during the test, just like in the monotonic bearing tests, as in Figure 5, even when loads were reduced to 70% of 60% bearing strength, (Table 2). The ovality for the plates that did not break was not extended, showing that plates with this stacking sequence are not suitable to tolerate deformations. In fact, fracture resulted after the displacement of the movable head of the test machine overcame 0.42 mm. This failure of some plates after a determined number of cycles made it possible to establish a preliminary correlation between load amplitude and the number of cycles to failure, as represented in Figure 8, resembling the well-known Wohler curve for metallic materials. Therefore, it is possible that, with a considerable number of tests, some correlation can be found and a “fatigue curve” can be drawn for these materials adapted to the diversity of the stacking sequences, making it possible to set a fatigue limit.

Finally, Figure 9 represents two examples of typical pin displacement versus the number of cycles observed in two different plates, one with fracture after 530 cycles, where an uprising of displacement is clearly visible prior to test stop, and another that has passed the 1000 cycles test without fracture.

An interesting feature noted in these tests was the higher on average cyclic resistance of holes drilled with HSS drills. This outcome may be the result of higher delamination in scattered directions, which acts as a barrier to fatigue failure progression. Based on the test results and the observation of the damage progression, it is evident that the progression of damage follows the same direction of the applied load, resulting in further cracks and increasing damage extension as the consecutive loads causes the breakage of more fibers. As the damage progresses, the resistant cross-section decreases due to the consecutive failure of fibers and matrix cracking and deformation, even though the matrix material is not mechanically strong but contributes to keep the bulk form of the plate. As the matrix material cracks, the bulk shape of the plate will easily be deformed as the fibers themselves cannot hold the original shape.

These assumptions must be confirmed with further studies and tests with a larger number of cycles, as is normal in fatigue studies.

### 3.4. Residual Bearing Strength Tests

The final step of this experimental study was the completion of bearing tests according to ASTM D5961 on the plates that supported 2000 cycles to determine the residual bearing resistance. So, three unidirectional and three cross-ply plates that were cyclic loaded and, consequently, deformed, as detailed in the precedent section, were tested and the results are presented in Table 6, including the bearing ratio, resulting from dividing the bearing strength after cyclic loading by the original average value for each drill condition on tool geometry.

For the unidirectional plates, there was no significative difference in the bearing strength due to previous deformation. In these plates, deformation seems to have had little or no effect on bearing strength. The decrease in the maximum bearing load is always small and the global average variation is less than 3%, demonstrating that unidirectional plates are not able to withstand deformation. On the other hand, for hole deformed cross-ply plates, the bearing strength decreased by around 9% if we consider Brad drilled plates, and by 7.5% on the global average of all valid test results. It is interesting to note that the wider the damage, the less the bearing strength decreases. In [28], the authors concluded a slight increase in bearing strength due to blunting. Notice that in [28], the number of cycles was 10^6^, much more than the number of cycles carried out in this study. So, it is reasonable to say that the blunting of the hole surrounding area can contribute to enhance fatigue resistance by hole distortion, which can be confirmed visually. All of these ratios need to be confirmed by performing a larger number of tests with a higher number of cycles. The rise in mechanical resistance to subsequent deformation due to hole distortion is a recognized reality in bearing strength tests. In ASTM D5961 standard [46], it is noted that it is possible to prevent masking of the true failure mode by large-scale hole distortion, as evidenced and confirmed within this study.

## 4. Conclusions

Here, we present an exploratory study on the conditions of damage propagation for fatigue testing on drilled composite plates in carbon/epoxy. The motivation was to gather a set of results for the future planning of long-term fatigue testing, which is time demanding and requires specialized equipment. For this purpose, two batches of carbon/epoxy plates, one with a unidirectional and the other with a cross-ply stacking sequence, were prepared for the above-described experimental sequence, including drilling, which is necessary for assembling purposes, and damage assessment by image processing, using existing damage criteria and low frequency cyclic testing, following ASTM D6873 procedure. From the work performed, some conclusions were made as follows:
By using enhanced radiography, both for delamination measurement and ovality progression, with the help of MatLab^®^ Image Processing tools, the delamination assessment procedure can assist in measuring damage progression;As the drilling process causes more damage around the hole, the bearing strength of the drilled plate decreases, evidencing the importance of proper cutting parameters to enhance reliability in service;As predicted, fatigue loading causes ovality of machined holes and decreases the bearing load capacity of machined holes after a certain number of cycles;The response to cyclic tests is different for the two stacking sequences of this work, showing that unidirectional plates are not suitable for load bearing or cyclic fatigue bearing loads. For cross-ply plates, it is possible to identify an asymptote of the fatigue vs. number of cycles curve.

Additionally, further studies are recommended based on these primary conclusions.

Cyclic tests need be performed with a larger number of cycles, from 20,000 to 2 × 10^6^ at 5 Hz frequency, with an adequate setup for image recording and interrupted at defined moments for radiography and ovality assessment;

An extended number of tests to study the possible effect of machining parameters on fatigue resistance, as well as stacking sequence effects, should be performed;

Another possible path of exploration is the use of hybrid composites with carbon and glass fibers as reinforcement.

## Figures and Tables

**Figure 1 materials-16-02688-f001:**
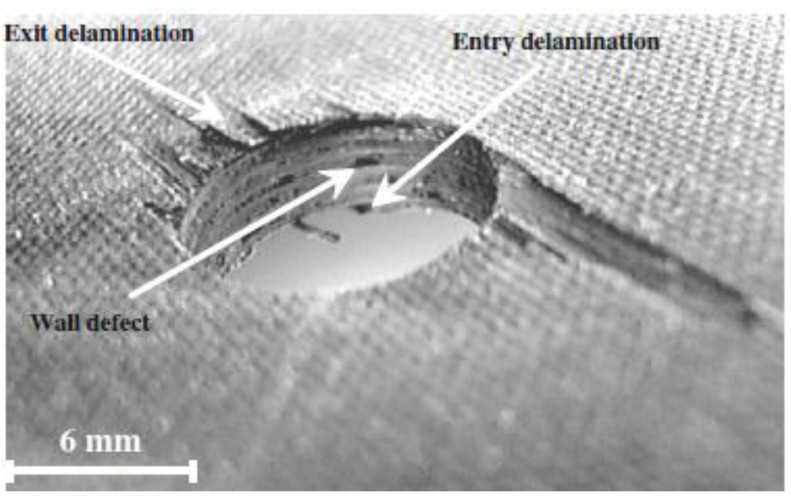
Drilling damages observed on CFRP laminate [1]. (Reprinted with permission from Ref. [1]. 2023, Springer Nature, https://www.springernature.com/gp).

**Figure 2 materials-16-02688-f002:**
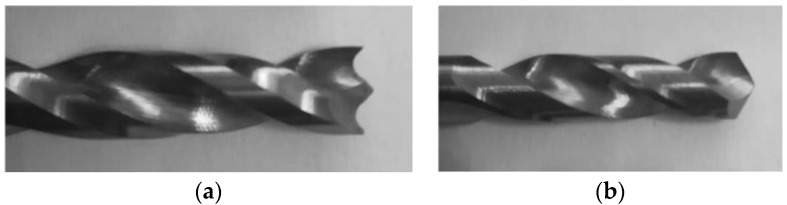
Drill geometries considered in experimental work: (**a**) Brad; (**b**) Twist.

**Figure 3 materials-16-02688-f003:**
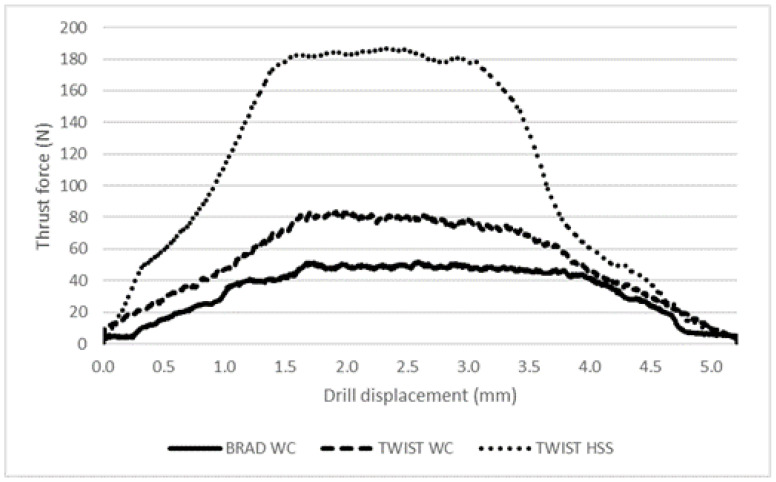
Comparison of thrust force during drilling of plates.

**Figure 4 materials-16-02688-f004:**
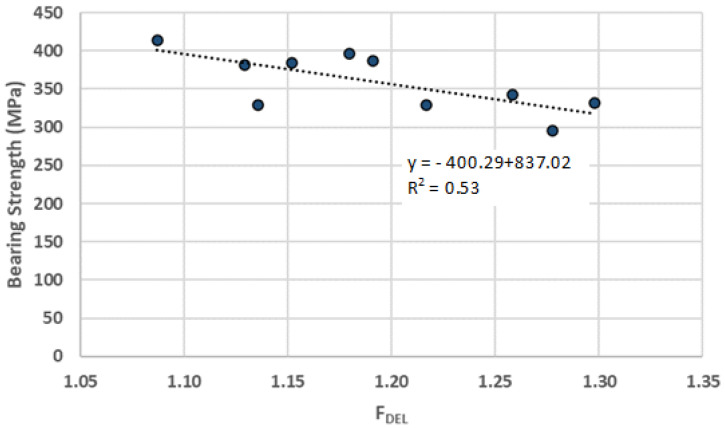
Correlation between Damage and Bearing strength for cross-ply plates.

**Figure 5 materials-16-02688-f005:**
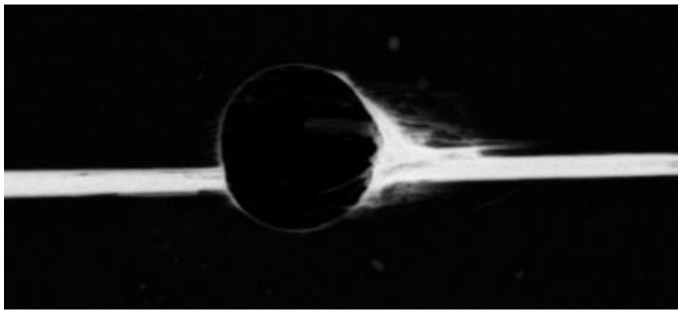
Unidirectional plate after Bearing test—the longitudinal slit is evident.

**Figure 6 materials-16-02688-f006:**
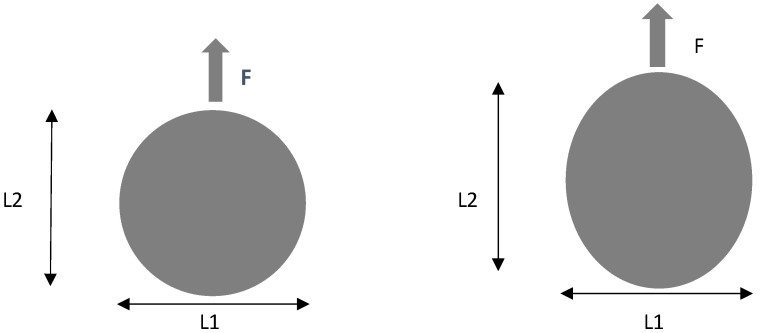
Ovality of drilled holes due to cyclic bearing test—concept.

**Figure 7 materials-16-02688-f007:**
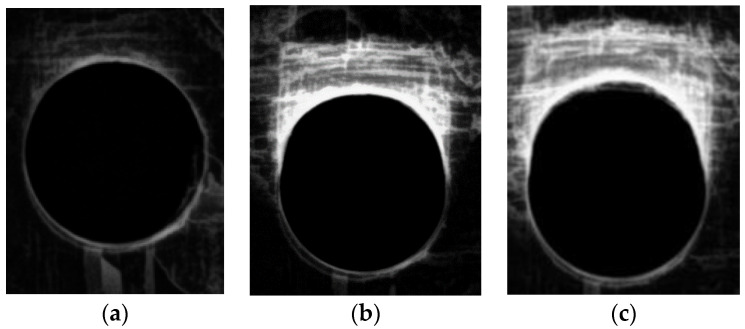
Radiography of drilled cross-ply plates: (**a**) as drilled; (**b**) after 1000 cycles; (**c**) after 2000 cycles—ovality in the load direction (top) is evident.

**Figure 8 materials-16-02688-f008:**
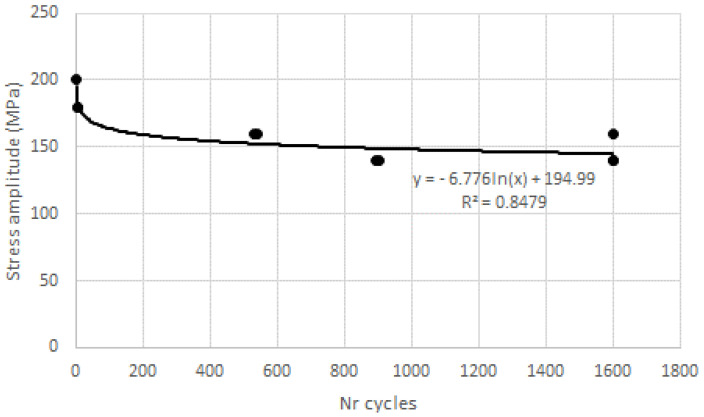
Stress amplitude vs. number of cycles for unidirectional plates.

**Figure 9 materials-16-02688-f009:**
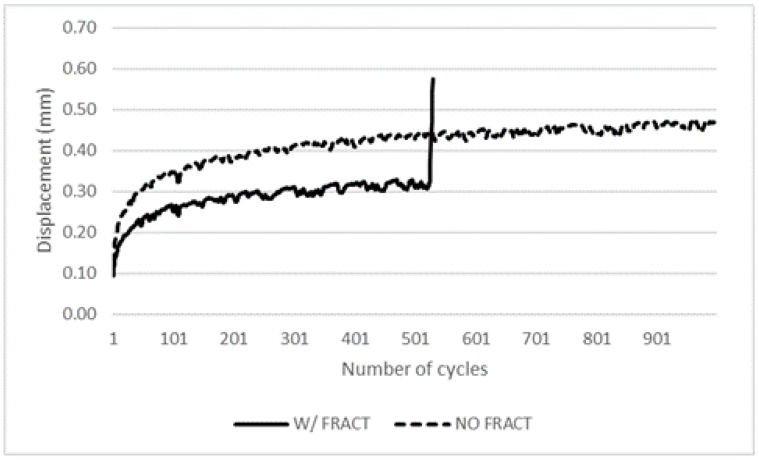
Pin displacement vs. number of cycles for cross-ply plates.

**Table 1 materials-16-02688-t001:** Mechanical properties of prepreg CIT HS160 T700 ER450 UD tape 36% [41].

Property	Experimental Value	Unit	Standard
Ultimate tensile strength	1700	MPa	ASTM D3039
Elastic modulus	111	GPa	ASTM D3039
Poisson coefficient	0.29	---	ASTM D3039
Elongation	1.72	%	ASTM D3039
Interlaminar fracture toughness in Mode I	419	N/m	ASTM D5528

**Table 2 materials-16-02688-t002:** Maximum bearing stresses in cyclic tests (all values in MPa).

LOADING	Reference Value	90%	80%	70%	60%
UNIDIRECTIONAL	205	185	164	144	123
CROSS PLY	360	324	288	252	216
OBSERVATIONS	ASTM D5961	ASTM D6873			

**Table 3 materials-16-02688-t003:** Thrust forces: maximum and at 0.15 mm from breakthrough (values in N).

DRILL GEOMETRY	MAX THRUST FORCE	THRUST FORCE AT 0.15 mm
BRAD WC	51	12.6
TWIST WC	87	12.5
TWIST HSS	183	22.1

**Table 4 materials-16-02688-t004:** Delamination factor (Equation (2)) and bearing strength of drilled plates.

PLATE	DRILL	DELAMINATION	BEARING STRENGTH (MPa)
UNIDIRECTIONAL	BRAD	1.12	191
	TWIST WC	1.12	210
	TWIST HSS	1.32	215
CROSS PLY	BRAD	1.18	386
	TWIST WC	1.12	380
	TWIST HSS	1.28	325

**Table 5 materials-16-02688-t005:** Ovality of drilled holes after bearing cycles ASTM D-6873 (values in mm).

Drilling Tool	L1	L2/as Drilled	L2/1000 Cycles	L2/2000 Cycles
WC BRAD	6.03	6.04	6.10	6.75
WC TWIST	5.95	6.06	6.14	6.66
HSS TWIST	5.92	5.97	6.17	6.71

**Table 6 materials-16-02688-t006:** Residual bearing strength of plates after cyclic testing.

PLATE	DRILL	DELAMINATION AFTER DRILLING	RESIDUAL BEARING STRENGTH (MPa)	BEARING RATIO
UNIDIRECTIONAL	BRAD	1.12	191	0.96
	TWIST WC	1.12	208	0.97
	TWIST HSS	1.32	202	0.98
CROSS PLY	BRAD	1.18	346	0.91
	TWIST WC	1.12	339	0.95
	TWIST HSS	1.28	314	0.97

## Data Availability

Research data is unavailable.

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
