# Peer review of "Damage Propagation by Cyclic Loading in Drilled Carbon/Epoxy Plates"

_materials, 2023, doi:10.3390/ma16072688_

Round 1

Reviewer 1 Report

Damage propagation by cyclic loading in drilled carbon/epoxy plates was investigated experimentally. Some research results obtained may be of significance to understand the damage of CFRP plates under the cyclic loading. The authors are encouraged to consider the following comments and make necessary improvements. 

1. The abstract should be rewritten, including more research results and conclusions of this paper, especially some quantitative analysis results and conclusions. In addition, the main work of this paper should be briefly described.

2. The writing of introduction should be improved through considering the following comments:

(1) When introducing the basic information of carbon fiber reinforced composite plates, the description of high impact resistance of CFRP is not accurate compared with the metallic alloys. This is due to the carbon fiber has low elongation and epoxy resin has the general brittle failure. In addition, the summaries of other performances and advantages should be verified by relevant research findings, such as high mechanical properties, excellent corrosion resistance, fatigue resistance, creep resistance. (2) The damage process and failure mechanism of CFRP under cyclic load should be summarized in detail, such as formation and propagation of cracks. The authors are encouraged to make necessary supplement by considering the above comments and relevant research work, such as Composite Structures, 2022, 293, 115719. Advances in Structural Engineering, 2022, 25(5): 939-953. Engineering Fracture Mechanics, 2022, 260: 108208. (3) In the last part, when introducing the research work, the authors further simplify the main contents of current research work, and at the same time, the innovations and contributions made by the research work should be highlighted and emphasized.

3. The clarity of many pictures in the current paper is very poor. It is recommended that the author check and revise all figures.

4. For the correlation between damage and bearing strength for cross-ply plates, please add relevant explanation and analysis of bearing strength decrease, such as crack size, formation and propagation.

5. What is the evolution mechanism of the properties of composite plates under cyclic loading? Please add relevant analysis and summary.

6. It is recommended to provide the ratio of bearing strength after cyclic testing as shown in Table 6. At the same time, the relevant analysis should be further enriched on the strength change.

7. The conclusion should be obtained based on the main results or findings in this paper. It should be distinguished from the future outlook.

Author Response

Dear reviewer 1, we would like to appreciate your effort and contribution to the enhancement of our manuscript. We have carefully addressed all your comments, performing a major revision of our work. Details of the changes are described in the following lines. 

  1. The abstract should be rewritten, including more research results and conclusions of this paper, especially some quantitative analysis results and conclusions. In addition, the main work of this paper should be briefly described.

REPLY: abstract was reviewed, including some highlights of the conclusions

2. The writing of introduction should be improved through considering the following comments:

(1) When introducing the basic information of carbon fiber reinforced composite plates, the description of high impact resistance of CFRP is not accurate compared with the metallic alloys. This is due to the carbon fiber has low elongation and epoxy resin has the general brittle failure. In addition, the summaries of other performances and advantages should be verified by relevant research findings, such as high mechanical properties, excellent corrosion resistance, fatigue resistance, creep resistance. (2) The damage process and failure mechanism of CFRP under cyclic load should be summarized in detail, such as formation and propagation of cracks. The authors are encouraged to make necessary supplement by considering the above comments and relevant research work, such as Composite Structures, 2022, 293, 115719. Advances in Structural Engineering, 2022, 25(5): 939-953. Engineering Fracture Mechanics, 2022, 260: 108208. (3) In the last part, when introducing the research work, the authors further simplify the main contents of current research work, and at the same time, the innovations and contributions made by the research work should be highlighted and emphasized. 

The introduction was largely re-written and new references were included. Regarding the text, we invite the reviewer to go for the "clean" version for better understanding of the whole text. New references are:

2. Soutis, C., Fibre reinforced composites in aircraft construction, Progress in Aerospace Science, 2005, pp 143-151.

3. Bhagwan D. Agarwal, B.D.; Broutman, L.J.; Chandrashekhara, K. Analysis and Performance of Fiber Composites, 4th Edition, Wyley, ISBN: 978-1-119-38998-9, October 2017.

4. Ahmad, H.; Markina, A.A.; Porotnikov, M.V.; Ahmad, F. A review of carbon fiber materials in automotive industry. IOP Conference Series: Materials Science and Engineering, 971, 2020, 0320112020.

5. Sreejith, R.S.R., Fiber reinforced composites for aerospace and sports applications. Woodhead Publishing Series in Composites Science and Engineering, 2021, pp 821-859.

6. Ghabezi, P.; Khoran, M. Optimization of Drilling Parameters in Composite Sandwich Structures (PVC Core), Indian J.Sci.Res. 2(1), 2014, 173-179.

7. Ghabezi, P.; Farahani, M.; Shahmirzaloo, A.; Ghorbani, H.; Harrison, N.M. Defect evaluation of the honeycomb structures formed during the drilling process, International Journal of Damage Mechanics Vol. 29(3), 2020, pp 454–466.

8. Yasar, N.K.; Korkmaz, M.E.; Günay, M. Investigation on hole quality of cutting conditions in drilling of CFRP composite. MATEC Web Conf. , 112, 2017, 01013.

9. Rajkumar, D.; Ranjithkumar, P.; Jenarthanan, M.P.; Sathiya Narayanan, C. Experimental investigation and analysis of factors influencing delamination and thrust force during drilling of carbon-fibre reinforced polymer composites. Pigment Resin Technol, 46, 2017, pp 507-524.

10. Jinyang Xu; Linfeng Li; Geier, N.; Davim, J.P.; Ming Chen, Experimental study of drilling behaviors and damage issues for woven GFRP composites using special drills, Journal of Materials Research and Technology, 2022, 21, pp 1256 – 1273.

11. Geng, D.; Zhenyu Shao, Y.L.; Zhenghui Lu, Jun Cai, Xun Li, Xinggang Jiang, Deyuan Zhang, Delamination formation, evaluation and suppression during drilling of composite laminates: A review Composite Structures, 2019, 216, pp 168-186.

33. Rui Guo, Guijun Xian, Chenggao Li, Bin Hong, Effect of fiber hybrid mode on the tension–tension fatigue performance for the pultruded carbon/glass fiber reinforced polymer composite rod, Engineering Fracture Mechanics, 260, 2022, 108208.

36. Hubert Debski, H.; Samborski, S.; Rozylo, P.; Wysmulski, P. Stability and Load-Carrying Capacity of Thin-Walled FRP Composite Z-Profiles under Eccentric Compression, Materials, 2020, 13, 2956.

37. Wysmulski, P. Non-linear analysis of the postbuckling behaviour of eccentrically compressed composite channel-section columns, Composite Structures, 305, 2023, 116446.

39. Durão, L.M.P.; Gonçalves, D.J.S.; Tavares, J.M.R.S.; de Albuquerque, V.H.C.; Marques, A.T. Comparative analysis of drills for composite laminates. Journal of Composite Materials, 46(14), 2011, pp 1649–1659.

40. Luís Miguel P. Durão, L.M.P.; Tavares, J.M.R.S.; de Albuquerque, V.H.C.; Marques, J.F.S.; Andrade, O.N.G. Drilling Damage in Composite Material, Materials 7, 2014, pp 3802-3819.

  1. The clarity of many pictures in the current paper is very poor. It is recommended that the author check and revise all figures.

REPLY: All the figures were reviewed with the exception of the radiography as we are limited by the equipment.

4. For the correlation between damage and bearing strength for cross-ply plates, please add relevant explanation and analysis of bearing strength decrease, such as crack size, formation and propagation.

REPLY: Comment was added ". Therefore, it is possible to affirm that a larger extension of the delaminated area, meaning a reduction of the consolidated resistant area in the vicinity of the hole, will cause a reduction of the bearing strength capacity of the plates. "

5. What is the evolution mechanism of the properties of composite plates under cyclic loading? Please add relevant analysis and summary.

REPLY: The following text was added "Based on the test results and the observation of the damage progression, it is evident that the progression of damage follows the same direction of the applied load, originating further cracks and damage extension increase as the consecutive loads causes the breakage of more fibres. As the damage progresses, the resistant cross-section decreases due to consecutive failure of fibres and matrix cracking and deformation, even though the matrix material is not mechanically strong but contributes to keep the bulk form of the plate. As the matrix material cracks, the bulk shape of the plate will easily be deformed as the fibres themselves cannot hold the original shape."

6. It is recommended to provide the ratio of bearing strength after cyclic testing as shown in Table 6. At the same time, the relevant analysis should be further enriched on the strength change.

REPLY: This was a fine outcome to add, so Table 6 was changed accordingly and the following changes had resulted on the text "

So, three unidirectional and three cross-ply plates that were cyclic loaded and, consequently, deformed as detailed in the precedent section, were tested and the results are presented in Table 6, including the bearing ratio, resulting from dividing the bearing strength after cyclic loading by the original average value for each drill condition on tool geometry.

For the unidirectional plates, there was no significative difference for the bearing strength due to previous deformation. In these plates, deformation seems to have little or no effect on bearing strength. The decrease of the maximum bearing load is always small, the global average variation is less than 3%, and it shows that unidirectional plates are not able to withstand deformation. On the other hand, for hole deformed cross-ply plates, the bearing strength has decreased by around 9%, if we consider Brad drilled plates and by 7.5% on the global average of all valid test results. It is interesting to note that the wider the damage the less is the bearing strength decrease. In [28], the authors had concluded by a slight increase of bearing strength due to blunting. Notice that in [28] the number of cycles was 106, much more than the number of cycles carried out in this work. So, it is possible to say that the blunting of the hole surrounding area can contribute to enhance fatigue resistance by hole distortion, which can be confirmed visually. All these ratios need to be confirmed by performing a larger number of tests with a higher number of cycles. The rise of mechanical resistance to subsequent deformation due to hole distortion is a recognized reality in bearing strength tests. In ASTM D5961 standard [46], there is a remark to prevent masking of the true failure mode by large-scale hole distortion, as evidenced and confirmed along this work."

7. The conclusion should be obtained based on the main results or findings in this paper. It should be distinguished from the future outlook.

REPLY: Thank you for your comment, we have separated main results from future outlook.

From the work performed, some conclusions are possible, as follows:

  • Delamination assessment procedure by using enhanced radiography, both for delamination measurement and ovality progression, with the help of MatLab® Image Processing tools can assist on damage progression measurement;
  • As the drilling process causes more damage around the hole, the bearing strength of the drilled plate decreases, evidencing the importance of proper cutting parameters to enhance reliability in service;
  • Fatigue loading causes ovality of machined holes and a decrease of the bearing load capacity of machined holes after a certain number of cycles as predicted;
  • The response to cyclic tests is different for the two stacking sequences of this work, showing that unidirectional plates are not suitable for load bearing or cyclic fatigue bearing loads. For cross-ply plates, it is possible to identify an asymptote of the fatigue vs number of cycles curve.

Also, some more work is recommended, based on these primary conclusions.

Cyclic tests need be performed with a larger number of cycles – from 20 000 to 2*106 at 5 Hz frequency – with an adequate setup for image recording and interrupted at defined moments for radiography and ovality assessment;

Extended number of tests to study the possible effect of machining parameters on fatigue resistance, as well as stacking sequence effects should be performed;

Another possibility to be explored is the use of hybrid composites – with carbon and glass fibres as reinforcement.

Reviewer 2 Report

The manuscript entitled: Damage Propagation by Cyclic Loading in drilled Carbon/ epoxy plates presents interesting research results.

The authors of this paper looked into the analysis of the drilling process in carbon epoxy plates. The surface area and other features of the damaged area are calculated. Finally, the samples were subjected to open-hole fatigue tests.

The research results described in the article correspond to the topics of the journal Materials. The following comments will help to improve the manuscript:

1) Add what method the composite panels were made using. Was it a vacuum package polymerised in an autoclave?

2) What is the exact layout of the laminate under test? In what direction is the first layer of 0 feel 90 oriented? The ply stack is symmetrical with respect to the midplane or the middle layer? e.g. [[0/90]6]s

3) Explain how the properties shown in Table 1 were determined or provide a reference.

4) Add figures and describe the experimental stand from the following description:

1)“The experimental work initiated with the drilling of the laminate plates for thrust force monitoring, delamination measurement by enhanced radiography and automated computational algorithms of image processing and analysis and, finally, mechanical tests.”

2) “The next step was the completion of the mechanical tests. In the first place, a few 204 coupons were tested according to ASTM D5961-17 [31]—Procedure A, both for unidirec-205 tional and cross-ply plates.”

5) In Figure 4, the description of the approximation function is incorrectly shown: y =

0,.394x + 0,.829

R² = 0,.001

Change this notation to y =

0.394x + 0.829

R² = 0.001

6) In my opinion, the introduction to the topic could be expanded. Manuscripts presenting experimental studies of carbon/epoxy composite laminates could be helpful. I suggest reading the following actual papers presenting studies of the damage of plate elements made of CFRP.

10.3390/ma13132956

10.1016/j.compstruct.2022.116446

Author Response

First of all, we would like to appreciate reviewer 2 for his contribution to the improvement of the manuscript. Several changes were implemented as result of comments, as it is possible to see in the attached file, with a clean version at the end of it. Regarding your comments, we add our replies.

1) Add what method the composite panels were made using. Was it a vacuum package polymerised in an autoclave?

R: In fact it was autoclave, we had added that information in the text "The test plates were produced from CIT (Composite Materials Italy, https://www.composite-materials.it/pagina.php?cod=1) carbon prepreg “CIT HS160 T700 ER450 UD tape 36%” and cured in autoclave for one hour under 300 kPa and 130 °C, followed by cooling"

2) What is the exact layout of the laminate under test? In what direction is the first layer of 0 feel 90 oriented? The ply stack is symmetrical with respect to the midplane or the middle layer? e.g. [[0/90]6]s

R: For us, the first layer is always 0º orientated. The relevant information was added "The cross-ply test plates consisted of 24 layers of unidirectional prepreg, stacked in a symmetric cross-ply sequence, alternating 90° between each layer, corresponding to a stacking sequence of [(0/90)6]s, "

3) Explain how the properties shown in Table 1 were determined or provide a reference.

R: Reference 41 was added, as the properties were in the technical sheet of the prepreg supplier

4) Add figures and describe the experimental stand from the following description:

1)“The experimental work initiated with the drilling of the laminate plates for thrust force monitoring, delamination measurement by enhanced radiography and automated computational algorithms of image processing and analysis and, finally, mechanical tests.”

2) “The next step was the completion of the mechanical tests. In the first place, a few coupons were tested according to ASTM D5961-17 [31]—Procedure A, both for unidirectional and cross-ply plates.”

R: The experimental sequence has been used before by the authors, so we opt to add some previous references to work [39, 40] and some text about it "

This experimental sequence has been presented in past papers from the same research team, see [39-40]. The only significant difference is the loading cycle, which is programmed with the help of the Trapezium software available with the universal testing machine."

5) In Figure 4, the description of the approximation function is incorrectly shown: y =

0,.394x + 0,.829

R² = 0,.001

Change this notation to y =

0.394x + 0.829

R² = 0.001

R: Yes, you are correct, that problem was missed in our first version, all figures were enhanced and equations and number formatting checked and corrected as there was also a mistake.

6) In my opinion, the introduction to the topic could be expanded. Manuscripts presenting experimental studies of carbon/epoxy composite laminates could be helpful. I suggest reading the following actual papers presenting studies of the damage of plate elements made of CFRP.

10.3390/ma13132956

10.1016/j.compstruct.2022.116446

R: The introduction was deeply changed and a number of references were added, as follows:

2. Soutis, C., Fibre reinforced composites in aircraft construction, Progress in Aerospace Science, 2005, pp 143-151.

3. Bhagwan D. Agarwal, B.D.; Broutman, L.J.; Chandrashekhara, K. Analysis and Performance of Fiber Composites, 4th Edition, Wyley, ISBN: 978-1-119-38998-9, October 2017.

4. Ahmad, H.; Markina, A.A.; Porotnikov, M.V.; Ahmad, F. A review of carbon fiber materials in automotive industry. IOP Conference Series: Materials Science and Engineering, 971, 2020, 0320112020.

5. Sreejith, R.S.R., Fiber reinforced composites for aerospace and sports applications. Woodhead Publishing Series in Composites Science and Engineering, 2021, pp 821-859.

6. Ghabezi, P.; Khoran, M. Optimization of Drilling Parameters in Composite Sandwich Structures (PVC Core), Indian J.Sci.Res. 2(1), 2014, 173-179.

7. Ghabezi, P.; Farahani, M.; Shahmirzaloo, A.; Ghorbani, H.; Harrison, N.M. Defect evaluation of the honeycomb structures formed during the drilling process, International Journal of Damage Mechanics Vol. 29(3), 2020, pp 454–466.

8. Yasar, N.K.; Korkmaz, M.E.; Günay, M. Investigation on hole quality of cutting conditions in drilling of CFRP composite. MATEC Web Conf. , 112, 2017, 01013.

9. Rajkumar, D.; Ranjithkumar, P.; Jenarthanan, M.P.; Sathiya Narayanan, C. Experimental investigation and analysis of factors influencing delamination and thrust force during drilling of carbon-fibre reinforced polymer composites. Pigment Resin Technol, 46, 2017, pp 507-524.

10. Jinyang Xu; Linfeng Li; Geier, N.; Davim, J.P.; Ming Chen, Experimental study of drilling behaviors and damage issues for woven GFRP composites using special drills, Journal of Materials Research and Technology, 2022, 21, pp 1256 – 1273.

11. Geng, D.; Zhenyu Shao, Y.L.; Zhenghui Lu, Jun Cai, Xun Li, Xinggang Jiang, Deyuan Zhang, Delamination formation, evaluation and suppression during drilling of composite laminates: A review Composite Structures, 2019, 216, pp 168-186.

33. Rui Guo, Guijun Xian, Chenggao Li, Bin Hong, Effect of fiber hybrid mode on the tension–tension fatigue performance for the pultruded carbon/glass fiber reinforced polymer composite rod, Engineering Fracture Mechanics, 260, 2022, 108208.

36. Hubert Debski, H.; Samborski, S.; Rozylo, P.; Wysmulski, P. Stability and Load-Carrying Capacity of Thin-Walled FRP Composite Z-Profiles under Eccentric Compression, Materials, 2020, 13, 2956.

37. Wysmulski, P. Non-linear analysis of the postbuckling behaviour of eccentrically compressed composite channel-section columns, Composite Structures, 305, 2023, 116446.

39. Durão, L.M.P.; Gonçalves, D.J.S.; Tavares, J.M.R.S.; de Albuquerque, V.H.C.; Marques, A.T. Comparative analysis of drills for composite laminates. Journal of Composite Materials, 46(14), 2011, pp 1649–1659.

40. Luís Miguel P. Durão, L.M.P.; Tavares, J.M.R.S.; de Albuquerque, V.H.C.; Marques, J.F.S.; Andrade, O.N.G. Drilling Damage in Composite Material, Materials 7, 2014, pp 3802-3819.

Reviewer 3 Report

The authors have investigated a batch of carbon/epoxy plates under different drilling conditions, and inspected hole quality. The paper is well-written, but it is necessary to summarize recently published papers in this area. Here are some comments to improve the quality of the work.

·       The abstract must be a single paragraph.

·       The abstract should cover a couple of main achievements of the work in its last sentences.

·       Page 2, line 47: Some of the references are too old, and there are some good recently published works in terms of drilling quality assessment (pull-out and Peel-up) in composite laminates such as the below ones and more recently published papers in this area. Review and summarize them to highlight the difference between your work and their achievements.

Optimization of drilling parameters in composite sandwich structures (PVC core)

Experimental study of drilling behaviors and damage issues for woven GFRP composites using special drills

Defect evaluation of the honeycomb structures formed during the drilling process

·       You need to add the most recently published papers in this area to show that this topic is still state-of-the-art.

·       The last paragraph of the introduction is similar to the last paragraph of the abstract.

·       The last paragraph of the introduction must present the novelty of your work.

·       More details regarding the used prepreg materials should be added including the name of the company, country, product code, etc.

·       Table 1 needs a reference.

·       The quality of figure 5 is not appropriate.

·       The main achievements of the work should be highlighted by bullets in the conclusion section.

Author Response

First, let us appreciate to reviewer 3 for his comments, has they help us to enhance our manuscript. Comments were addressed and the answers can be found in the text below and on the revised manuscript.

The abstract must be a single paragraph.

The abstract should cover a couple of main achievements of the work in its last sentences.

R: Abstract was rewritten accordingly.

Page 2, line 47: Some of the references are too old, and there are some good recently published works in terms of drilling quality assessment (pull-out and Peel-up) in composite laminates such as the below ones and more recently published papers in this area. Review and summarize them to highlight the difference between your work and their achievements.

Optimization of drilling parameters in composite sandwich structures (PVC core)

Experimental study of drilling behaviors and damage issues for woven GFRP composites using special drills

Defect evaluation of the honeycomb structures formed during the drilling process

R: Although the core of these published papers is not the same as ours, we made our best to include them in the revised manuscript.

You need to add the most recently published papers in this area to show that this topic is still state-of-the-art.

R: New references were added to the text, as follows

2. Soutis, C., Fibre reinforced composites in aircraft construction, Progress in Aerospace Science, 2005, pp 143-151.

3. Bhagwan D. Agarwal, B.D.; Broutman, L.J.; Chandrashekhara, K. Analysis and Performance of Fiber Composites, 4th Edition, Wyley, ISBN: 978-1-119-38998-9, October 2017.

4. Ahmad, H.; Markina, A.A.; Porotnikov, M.V.; Ahmad, F. A review of carbon fiber materials in automotive industry. IOP Conference Series: Materials Science and Engineering, 971, 2020, 0320112020.

5. Sreejith, R.S.R., Fiber reinforced composites for aerospace and sports applications. Woodhead Publishing Series in Composites Science and Engineering, 2021, pp 821-859.

6. Ghabezi, P.; Khoran, M. Optimization of Drilling Parameters in Composite Sandwich Structures (PVC Core), Indian J.Sci.Res. 2(1), 2014, 173-179.

7. Ghabezi, P.; Farahani, M.; Shahmirzaloo, A.; Ghorbani, H.; Harrison, N.M. Defect evaluation of the honeycomb structures formed during the drilling process, International Journal of Damage Mechanics Vol. 29(3), 2020, pp 454–466.

8. Yasar, N.K.; Korkmaz, M.E.; Günay, M. Investigation on hole quality of cutting conditions in drilling of CFRP composite. MATEC Web Conf. , 112, 2017, 01013.

9. Rajkumar, D.; Ranjithkumar, P.; Jenarthanan, M.P.; Sathiya Narayanan, C. Experimental investigation and analysis of factors influencing delamination and thrust force during drilling of carbon-fibre reinforced polymer composites. Pigment Resin Technol, 46, 2017, pp 507-524.

10. Jinyang Xu; Linfeng Li; Geier, N.; Davim, J.P.; Ming Chen, Experimental study of drilling behaviors and damage issues for woven GFRP composites using special drills, Journal of Materials Research and Technology, 2022, 21, pp 1256 – 1273.

11. Geng, D.; Zhenyu Shao, Y.L.; Zhenghui Lu, Jun Cai, Xun Li, Xinggang Jiang, Deyuan Zhang, Delamination formation, evaluation and suppression during drilling of composite laminates: A review Composite Structures, 2019, 216, pp 168-186.

33. Rui Guo, Guijun Xian, Chenggao Li, Bin Hong, Effect of fiber hybrid mode on the tension–tension fatigue performance for the pultruded carbon/glass fiber reinforced polymer composite rod, Engineering Fracture Mechanics, 260, 2022, 108208.

36. Hubert Debski, H.; Samborski, S.; Rozylo, P.; Wysmulski, P. Stability and Load-Carrying Capacity of Thin-Walled FRP Composite Z-Profiles under Eccentric Compression, Materials, 2020, 13, 2956.

37. Wysmulski, P. Non-linear analysis of the postbuckling behaviour of ecentrically compressed composite channel-section columns, Composite Structures, 305, 2023, 116446.

39. Durão, L.M.P.; Gonçalves, D.J.S.; Tavares, J.M.R.S.; de Albuquerque, V.H.C.; Marques, A.T. Comparative analysis of drills for composite laminates. Journal of Composite Materials, 46(14), 2011, pp 1649–1659.

40. Luís Miguel P. Durão, L.M.P.; Tavares, J.M.R.S.; de Albuquerque, V.H.C.; Marques, J.F.S.; Andrade, O.N.G. Drilling Damage in Composite Material, Materials 7, 2014, pp 3802-3819.

The last paragraph of the introduction is similar to the last paragraph of the abstract.

R: Text was changed, as we completely agree with this observation

The results of the experimental sequence here presented allow to identify some differences in relation to the stacking sequences when under cyclic loads and, also, to have some information on the effects of cyclic loading on drilled composite plates. Whereas cross-ply plates were able to withstand some deformation, showing some ovality of the hole, most of the unidirectional plates have failed after a reduced number of cycles. For the cross-ply plates it was possible to draw a fatigue curve showing that a correlation between number of cycles and load amplitude can eventually be established. There is a decrease in the load bearing capacity of all the plates as the number of loading cycles increase. On the other hand, the load amplitude of the tests to unidirectional plates was always inferior to that of the cross-ply plates.

The last paragraph of the introduction must present the novelty of your work.

R: Please see our answer on the question above.

More details regarding the used prepreg materials should be added including the name of the company, country, product code, etc.

R: The mentioned information was added to the text

The test plates were produced from CIT (Composite Materials Italy, https://www.composite-materials.it/pagina.php?cod=1) carbon prepreg “CIT HS160 T700 ER450 UD tape 36%” and cured in autoclave for one hour under 300 kPa and 130 °C, followed by cooling

Table 1 needs a reference.

R: Reference 41 was added to table 1

41. Composite Materials Italy, https://www.composite-materials.it/pagina.php?cod=1

The quality of figure 5 is not appropriate.

R: All figures were enhanced, except radiography, due to equipment constraints

The main achievements of the work should be highlighted by bullets in the conclusion section.

R: The conclusions section was changed

  • Delamination assessment procedure by using enhanced radiography, both for delamination measurement and ovality progression, with the help of MatLab® Image Processing tools can assist on damage progression measurement;
  • As the drilling process causes more damage around the hole, the bearing strength of the drilled plate decreases, evidencing the importance of proper cutting parameters to enhance reliability in service;
  • Fatigue loading causes ovality of machined holes and a decrease of the bearing load capacity of machined holes after a certain number of cycles as predicted;
  • The response to cyclic tests is different for the two stacking sequences of this work, showing that unidirectional plates are not suitable for load bearing or cyclic fatigue bearing loads. For cross-ply plates, it is possible to identify an asymptote of the fatigue vs number of cycles curve.

Also, some more work is recommended, based on these primary conclusions.

Cyclic tests need be performed with a larger number of cycles – from 20 000 to 2*106 at 5 Hz frequency – with an adequate setup for image recording and interrupted at defined moments for radiography and ovality assessment;

Extended number of tests to study the possible effect of machining parameters on fatigue resistance, as well as stacking sequence effects should be performed;

Another possibility to be explored is the use of hybrid composites – with carbon and glass fibres as reinforcement.

Round 2

Reviewer 1 Report

It can be accepted in the present form.

Reviewer 2 Report

Accept in present form